# Ensembling Graph Predictions for AMR Parsing

**Hoang Thanh Lam**[1], **Gabriele Picco**[1], **Yufang Hou**[1], **Young-Suk Lee**[2],
**Lam M. Nguyen**[2], **Dzung T. Phan**[2], **Vanessa López**[1], **Ramon Fernandez Astudillo**[2]

[1] IBM Research, Dublin, Ireland
[2] IBM Research, Thomas J. Watson Research Center, Yorktown Heights, USA
t.l.hoang@ie.ibm.com, gabriele.picco@ibm.com, yhou@ie.ibm.com,
ysuklee@us.ibm.com, LamNguyen.MLTD@ibm.com, phandu@us.ibm.com,
vanlopez@ie.ibm.com, ramon.astudillo@ibm.com

## Abstract

In many machine learning tasks, models are trained to predict structure data such as graphs. For example, in natural language processing, it is very common to parse texts into dependency trees or abstract meaning representation (AMR) graphs. On the other hand, ensemble methods combine predictions from multiple models to create a new one that is more robust and accurate than individual predictions. In the literature, there are many ensembling techniques proposed for classification or regression problems, however, ensemble graph prediction has not been studied thoroughly. In this work, we formalize this problem as mining the largest graph that is the most supported by a collection of graph predictions. As the problem is NP-Hard, we propose an efficient heuristic algorithm to approximate the optimal solution. To validate our approach, we carried out experiments in AMR parsing problems. The experimental results demonstrate that the proposed approach can combine the strength of state-of-the-art AMR parsers to create new predictions that are more accurate than any individual models in five standard benchmark datasets.

## 1 Introduction

Ensemble learning is a popular machine learning practice, in which predictions from multiple models are blended to create a new one that is usually more robust and accurate. Indeed, ensemble methods like XGBOOST are the winning solution in many machine learning and data science competitions [Chen and Guestrin, 2016]. A key reason behind the successes of the ensemble methods is that they can combine the strength of different models to reduce the variance and bias in the final prediction [Domingos, 2000, Valentini and Dietterich, 2004]. Research in ensemble methods mostly focuses on regression or classification problems [Dong et al., 2020]. Recently, in many machine learning tasks prediction outputs are provided in a form of graphs. For instance, in Abstract Meaning Representation (AMR) parsing [Banarescu et al., 2013], the input is a fragment of text and the output is a rooted, labeled, directed, acyclic graph (DAG). It abstracts away from syntactic representations, in the sense that sentences with similar meaning should have the same AMR. Figure 1 shows an AMR graph for the sentence *You told me to wash the dog* where nodes are concepts and edges are relations.

AMR parsing is an important problem in natural language processing (NLP) research and it has a broad application in downstream tasks such as question answering [Kapanipathi et al., 2020] and common sense reasoning [Lim et al., 2020]. Recent approaches for AMR parsing leverage the advances from pretrained language models [Bevilacqua et al., 2021] and numerous deep neural network architectures [Cai and Lam, 2020a, Zhou et al., 2021].

Unlike methods for ensembling numerical or categorical values for regression or classification problems where the mean value or majority votes are used respectively, the problem of graph ensemble is more complicated. For instance, Figure 2 show three graphs $g_1, g_2, g_3$ with different

35th Conference on Neural Information Processing Systems (NeurIPS 2021).

structures, having varied number of edges and vertices with different labels. In this work, we formulate the ensemble graph prediction as a graph mining problem where we look for the largest common structure among the graph predictions. In general, finding the largest common subgraph is a well-known computationally intractable problem in graph theory. However, for AMR parsing problems where the AMR graphs have labels and a simple tree-alike structure, we propose an efficient heuristic algorithm (*Graphene*) to approximate the solution of the given problem well.

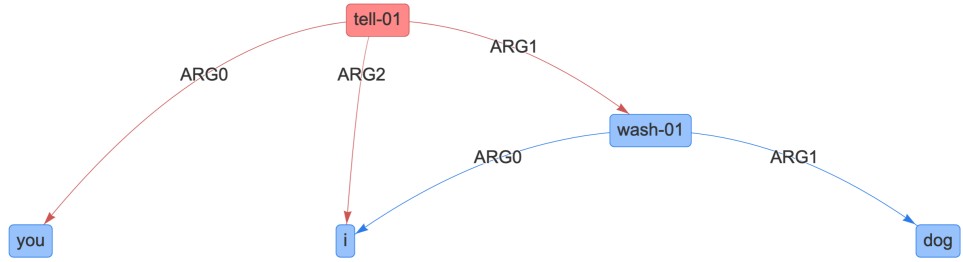

Figure 1: An example AMR graph for the sentence *You told me to wash the dog*.

To validate our approach, we collect the predictions from four state-of-the-art AMR parsers and create new predictions using the proposed graph ensemble algorithm. The chosen AMR parsers are the recent state-of-the-art AMR parsers including a seq2seq-based method using BART [Bevilacqua et al., 2021], a transition-based approach proposed in [Zhou et al., 2021] and a graph-based approach proposed in [Cai and Lam, 2020a]. In addition to those models, we also trained a new seq2seq model based on T5 [Raffel et al., 2020] to leverage the strength of this pretrained language model.

The experimental results show that in five standard benchmark datasets, our proposed ensemble approach outperforms the previous state-of-the-art models and achieves new state-of-the-art results in all datasets. For example, our approach achieves new state-of-the-art results with 1.7, 1.5, and 1.3 points better than prior arts in the BIO (under out-of-distribution evaluation), AMR 2.0, and AMR 3.0 datasets respectively. This result demonstrates the strength of our ensemble method in leveraging the model diversity to achieve better performance. An interesting property of our solution is that it is model-agnostic, therefore it can be used to make an ensemble of existing model predictions without the requirement to have an access to model training. Source code is open-sourced[1].

Our paper is organized as follows: Section 2 discusses a formal problem definition and a study on the computational intractability of the formulated problem. The graph ensemble algorithm is described in Section 3. Experimental results are reported in Section 4 while Section 5 discusses related works. The conclusion and future work are discussed in Section 6.

## 2  Problem formulation

Denote $g = (E, V)$ as a graph with the set of edges $E$ and the set of vertices $V$. Each vertex $v \in V$ and edge $e \in E$ is associated with a label denoted as $l(v)$ and $l(e)$ respectively, where $l(.)$ is a labelling function. Given two graphs $g_1 = (E_1, V_1)$ and $g_2 = (E_2, V_2)$, a *vertex matching* $\phi$ is a bijective function that maps a vertex $v \in V_1$ to a vertex $\phi(v) \in V_2$.

**Example 1.** *In Figure 2, between $g_1$ and $g_2$ there are many possible vertex matches, where $\phi(g_1, g_2) = [1 \rightarrow 3, 2 \rightarrow 2, 3 \rightarrow 1]$ is one of them (which can be read as the first vertex of $g_1$ is mapped to the third vertex of $g_2$ and so forth). Notice that not all vertices $v \in V_1$ has a match in $V_2$ and vice versa. Indeed, in this example, the fourth vertex in $g_2$ does not have a matched vertex in $g_1$.*

Given two graphs $g_1$, $g_2$ and a vertex match $\phi(g_1, g_2)$, support of a vertex $v$ with respect to the matching $\phi$, denoted as $s_\phi(v)$, is equal to 1 if $l(v) = l(\phi(v))$ and 0 otherwise. Given an edge $e = (v_1, v_2)$ the support of $e$ with respect to the vertex match $\phi$, denoted as $s_\phi(e)$, is equal to 1 if $l(e) = l((\phi(v_1), \phi(v_2)))$ and 0 otherwise.

---

[1] https://github.com/IBM/graph_ensemble_learning

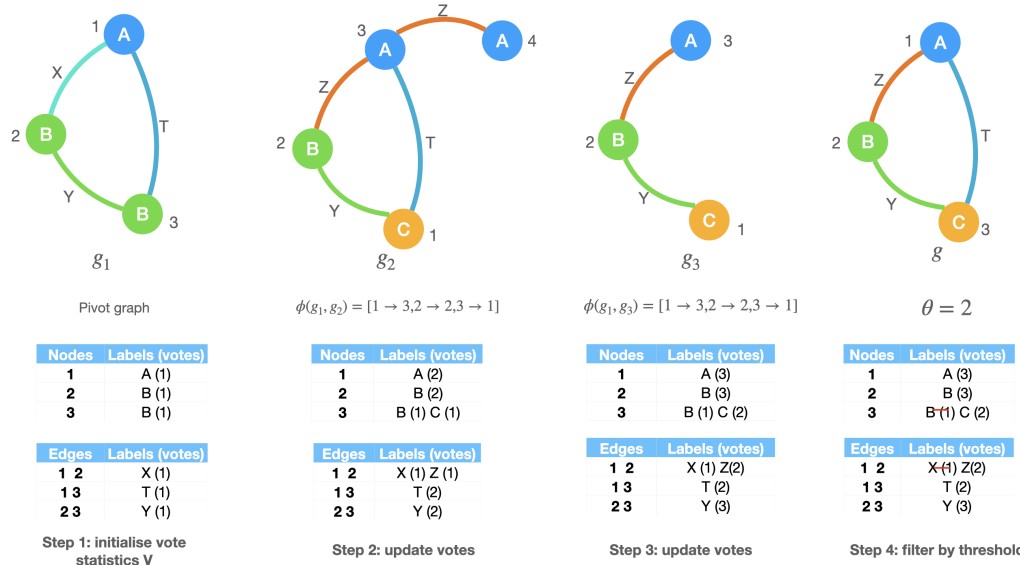

Figure 2: A graph ensemble example. Each node and edge of $g$ occurs in at least two out of three graphs $g_1, g_2, g_3$. Therefore, $g$ is $\theta$-supported where $\theta = 2$ by the given set of graphs. Graph $g$ is also the graph with the largest sum of supports among all $\theta$-supported graphs. The tables show the node and edge support (votes) are updated in each step of the Graphene algorithm when $g_1$ is a pivot graph.

**Example 2.** *In Figure 2, for the vertex match $\phi(g_1, g_2) = [1 \to 3, 2 \to 2, 3 \to 1]$, the first vertex in $g_1$ and the third vertex in $g_2$ shares the same label A, therefore the support of the given vertex is equal to 1. On the other hand, the third vertex in $g_1$ and the first vertex in $g_2$ does not have the same label so their support is equal to 0.*

Between two graphs, there are many possible vertex matches, the *best vertex match* is defined as the one that has the maximal total vertex support and edge support. In our discussion, when we mention a vertex match we always refer to the best vertex match.

Denote $G = \{g_1 = (E_1, V_1), g_2 = (E_2, V_2), \cdots, g_m = (E_m, V_m)\}$ as a set of $m$ graphs. Given any graph $g = (E, V)$, for every $g_i$ denote $\phi_i(g, g_i)$ as the best vertex match between $g$ and $g_i$. The total support of a vertex $v \in V$ or an edge $e \in E$ is defined as follows:

- $\text{support}(e) = \sum_{i=1}^{m} s_{\phi_i}(e)$
- $\text{support}(v) = \sum_{i=1}^{m} s_{\phi_i}(v)$

Given a support threshold $\theta$, a graph $g$ is called $\theta$-supported by $G$ if for any node $v \in V$ or any edge $e \in E$, $\text{support}(v) \geq \theta$ and $\text{support}(e) \geq \theta$.

**Example 3.** *In Figure 2, graph $g$ is $\theta$-supported by $G = \{g_1, g_2, g_3\}$ where $\theta = 2$.*

Intuitively, an ensemble graph $g$ should have as many common edges and vertices with all the graph predictions as possible. Therefore, we define the graph ensemble problem as follows:

**Problem 1** (Graph ensemble). *Given a support threshold $\theta$ and a collection of graphs G, find the graph $g$ that is $\theta$-supported by G and has the largest sum of vertex and edge supports.*

**Theorem 1.** *Finding the optimal $\theta$-supported graph with the largest total of support is NP-Hard.*

*Proof.* We prove the NP-Hardness by reduction to the *Maximum Common Edge Subgraph* (MCES) problem, which is known to be an NP-Complete problem [Bahiense et al., 2012]. Given two graphs $g_1$ and $g_2$, the MCES problem finds a graph $g$ that is a common subgraph of $g_1$ and $g_2$ and the number of edges in $g$ is the largest. Consider the following instance of the Graph Ensemble problem with $\theta = 2$, and $G = \{g_1, g_2\}$ created from the graphs in the MCES problem. Assume that all vertices and all edges of $g_1$ and $g_2$ have the same label $A$.

Since $\theta = 2$, a $\theta$-supported graph is also a common subgraph between $g_1$ and $g_2$ and vice versa. Denote $g_s$ and $g_e$ as the common subgraph between $g_1$ and $g_2$ with the largest support and the largest common edge, respectively. We can show that $g_s$ has as many edges as $g_e$. In fact, since $g_s$ is the largest supported common subgraph there is no vertex $v \in g_e$ such that $v \notin g_s$ because otherwise we can add $v$ to $g_s$ to create a larger supported graph. For any edge $e = (v_1, v_2) \in g_e$, since both vertices $v_1$ and $v_2$ also appear in $g_s$, the edge $e = (v_1, v_2)$ must also be part of $g_s$ otherwise we can add this edge to $g_s$ to create a subgraph with a larger support. Therefore, $g_s$ has as many edges as $g_e$, which is also a solution to the MCES problem. $\square$

## 3 Graph ensemble algorithm

In this section, we discuss a heuristic algorithm based on the strategy *"Please correct me if I am wrong!"* to solve Problem 1. The main idea is to improve a pivot graph based on other graphs. Specifically, starting with a pivot graph $g_i$ ($i = 1, 2, \cdots, m$), we collect the votes from the other graphs for every existing vertex and existing/non-existing edges to correct $g_i$. We call the proposed algorithm *Graphene* which stands for Graph Ensemble algorithm. The key steps of the algorithm are provided in the pseudo-code in Algorithm 1.

---

**Algorithm 1:** Graph ensemble with the Graphene algorithm.

---

**Input**: a set of graphs $G = \{g_1, g_2, \cdots, g_m\}$ and the support threshold $\theta$
**Output**: an ensemble graph $g^e$
**Algorithm**: Graphene$(G, \theta)$
**for** $i \leftarrow 1$ **to** $m$ **do**
    $g_{\text{pivot}} \leftarrow g_i$
    $V \leftarrow$ Initialise$(g_{\text{pivot}})$
    **for** $j \leftarrow 1$ **to** $m$ **do**
        **if** $j \neq i$ **then**
            $V \leftarrow V \cup$ getVote$(\phi(g_{\text{pivot}}, g_j))$
    **end**
    $g_i^e \leftarrow Filter(V, \theta)$
**end**
$g^e \leftarrow$ the graph with the largest support among $g_1^e, \cdots, g_m^e$
**Return** $g^e$

---

For example, in Figure 2, the algorithm starts with the first graph $g_1$ and considers it as a pivot graph $g_{\text{pivot}}$. In the first step, it creates a table to keep voting statistics $V$ initialized with the vote counts for every existing vertex and edge in $g_{\text{pivot}}$. To draw additional votes from the other graphs, it performs the following subsequent steps:

- Call the function $\phi(g_1, g_i)$ ($i = 2, 3, \cdots, m$) to get the best bijective mapping $\phi$ between the vertices of two graphs $g_1$ and $g_i$ (with a little bit abuse of notation we drop the index $i$ from $\phi_i$ when $g_i$ and $g_{\text{pivot}}$ are given in the context). For instance, the best vertex match between $g_1$ and $g_2$ is $\phi = 1 \rightarrow 3, 2 \rightarrow 2, 3 \rightarrow 1$ because that vertex match has the largest number of common labeled edges and vertices.

- Enumerate the matching vertices and edges to update the voting statistics accordingly. For instance, since the vertex 3 in $g_1$ with label $B$ is mapped to the vertex 1 in $g_2$ with label $C$, a new candidate label $C$ is added to the table for the given vertex. For the same reason, we add a new candidate label $Z$ for the edge $(1, 2)$. For all the other edges and vertices where the labels are matched the votes are updated accordingly.

Once the complete voting statistics $V$ is available, the algorithm filters the candidate labels of edges and vertices using the provided support threshold $\theta$ by calling the function $Filter(V, \theta)$ to obtain an ensemble graph $g_i^e$. For special cases, when disconnected graphs are not considered as a valid output, we keep all edges of the pivot graph even its support is below the threshold. On the other hand, for the graph prediction problem, where a graph is only considered a valid graph if it does not have multiple edges between two vertices and multiple labels for any vertex, we remove all candidate labels for vertices and edges except the one with the highest number of votes.

Assume that the resulting ensemble graph that is created by using $g_i$ as the pivot graph is denoted as $g_i^e$. The final ensemble graph $g^e$ is chosen among the set of graphs $g_1^e, g_2^e, \cdots, g_m^e$ as the one with the largest total support. Recall that $\phi(g_{\text{pivot}}, g_i)$ finds the best vertex match between two graphs. In general, the given task is computationally intractable. However, for labeled graphs like AMR a heuristic was proposed [Cai and Knight, 2013] to approximate the best match by a hill-climbing algorithm. It first starts with the candidate with labels that are mostly matched. The initial match is modified iteratively to optimize the total number of matches with a predefined number of iterations (default value set to 5). This algorithm is very efficient and effective, it was used to calculate the Smatch score in [Cai and Knight, 2013] so we reuse the same implementation to approximate $\phi(g_{\text{pivot}}, g_i)$ (report on average running time can be found in the supplementary materials).

## 4   Experiments

We compare our *Graphene* algorithm against four previous state-of-the-art models on different benchmark datasets. Below we describe our experimental settings.

### 4.1   Experimental settings

#### 4.1.1   Model settings

**SPRING**   The SPRING model, presented in [Bevilacqua et al., 2021], tackles Text-to-AMR and AMR-to-Text as a symmetric transduction task. The authors show that with a pretrained encoder-decoder model, it is possible to obtain state-of-the-art performances in both tasks using a simple seq2seq framework by predicting linearized graphs. In our experiments, we used the pretrained models provided in [Bevilacqua et al., 2021][2]. In addition, we trained 3 more models using different random seeds following the same setup described in [Bevilacqua et al., 2021]. Blink [Li et al., 2020] was used to add wiki tags to the predicted AMR graphs as a post-processing step.

**T5**   The T5 model, presented in [Raffel et al., 2020], introduces a unified framework that models a wide range of NLP tasks as a text-to-text problem. We follow the same idea proposed in [Xu et al., 2020] to train a model to transfer a text to a linearized AMR graph based on T5-Large. The data is preprocessed by linearization and removing wiki tags using the script provided in [amr]. In addition to the main task, we added a new task that takes as input a sentence and predicts the concatenation of word senses and arguments provided in the English Web Treebank dataset [goo]. The model is trained with 30 epochs. We use ADAM optimization with a learning rate of 1e-4 and a mini-batch size of 4. Blink [Li et al., 2020] was used to add wiki tags to the predicted AMR graphs during post-processing.

**APT**   [Zhou et al., 2021] proposed a transition-based AMR parser[3] based on Transformer [Vaswani et al., 2017]. It combines hard-attentions over sentences with a target side action pointer mechanism to decouple source tokens from node representations. In our experiments, we use the setup described in [Zhou et al., 2021] and added 70K model-annotated silver sentences to the training data, which was created from the 85K sentence set in [Lee et al., 2020] with self-learning described in the paper.

**Cai&Lam**   The model proposed in [Cai and Lam, 2020b] treats AMR parsing as a series of dual decisions (i.e., *which parts of the sequence to abstract*, and *where in the graph to construct*) on the input sequence and constructs the AMR graph incrementally. Following [Cai and Lam, 2020b], we use Stanford CoreNLP[4] for tokenization, lemmatization, part-of-speech tagging, and named entity recognition. We apply the pretrained model provided by the authors[5] to all testing datasets and follow the same pre-processing and post-processing steps for graph re-categorization.

**Graphene (our algorithm)**   The only hyperparameter of the Graphene algorithm is the threshold $\theta$. Following the majority voting strategy [Dong et al., 2020], we set the threshold $\theta$ such that $\frac{\theta}{m} \geq 0.5$ (where $m$ is the number of models in the ensemble). In all experiments, we used a Tesla GPU V100 for model training and used 8 CPUs for making an ensemble.

---

[2] Available for download at `https://github.com/SapienzaNLP/spring`

[3] Available under https://github.com/IBM/ transition-amr-parser.

[4] Available at `https://github.com/stanfordnlp/stanza/`

[5] The model "AMR2.0+BERT+GR" can be downloaded from `https://github.com/jcyk/AMR-gs`

### 4.1.2 Evaluation

We use the script[6] provided in [Damonte et al., 2017] to calculate the Smatch score [Cai and Knight, 2013], the most relevant metric for measuring the similarity between the predictions and the gold AMR graphs. The overall Smatch score can be broken down into different sub-metrics:

- Unlabeled (Unl.): Smatch score after removing all edge labels
- No WSD (NWSD): Smatch score while ignoring Propbank senses.
- NE: F-score on the named entity recognition (:name roles)
- Wikification (Wiki.): F-score on the wikification (:wiki roles)
- Negations (Neg.): F-score on the negation detection (:polarity roles)
- Concepts (Con.): F-score on the concept identification task
- Reentrancy (Reen.): Smatch computed on reentrant edges only
- SRL: Smatch computed on :ARG-i roles only

### 4.1.3 Datasets

Similarly to [Bevilacqua et al., 2021], we use five standard benchmark datasets [dat] to evaluate our approach. Table 1 shows the statistics of the datasets. AMR 2.0 and AMR 3.0 are divided into train, development and testing sets and we use them for *in-distribution* evaluation in Section 4.2. Furthermore, the models trained on AMR 2.0 training data are used to evaluate *out-of-distribution* prediction on the BIO, the LP and the New3 dataset (See Section 4.3).

Table 1: Benchmark datasets. All instances of BIO, LP, and New3 are used to test models in out-of-distribution evaluation. For AMR 2.0 and 3.0, the models are trained on the training dataset, validated on the development dataset. We report results on testing sets in the in-distribution evaluation.

| Datasets | AMR 2.0 | AMR 3.0 | BIO | Little Prince (LP) | New3 |
|---|---|---|---|---|---|
| Training | 36,521 | 55,635 | n/a | n/a | n/a |
| Dev | 1,368 | 1,722 | n/a | n/a | n/a |
| Test | 1,371 | 1,898 | 6,952 | 1,562 | 527 |

### 4.2 In-distribution evaluation

In the same spirit of [Bevilacqua et al., 2021], we evaluate the approaches when training and test data belong to the same domain. Table 2 shows the results of the models on the test split of the AMR 2.0 and AMR 3.0 datasets. The metrics reported for SPRING correspond to the model with the highest Smatch score among the 4 models(the checkpoint plus the 3 models with different random seeds).

For the ensemble approach, we report the result when Graphene is an ensemble of four SPRING checkpoints, denoted as *Graphene 4S*. The ensemble of all the models including four SPRING checkpoints, APT, T5, and Cai&Lam is denoted as *Graphene All*. For the AMR 3.0 dataset, the Cai&Lam model is not available so the reported result corresponds to an ensemble of all six models.

We can see that Graphene successfully leverages the strength of all the models and provides better prediction both in terms of the overall Smatch score and sub-metrics. In both datasets, we achieve the state-of-the-art results with performance gain of 1.6 and 1.2 Smatch points in AMR 2.0 and AMR 3.0 respectively. Table 2 shows that by combining predictions from four checkpoints of the SPRING model, Graphene 4S provides better results than any individual models. The result is improved further when increasing the number of ensemble models, indeed *Graphene All* improves *Graphene 4S* further and outperforms the individual models in terms of the overall Smatch score.

### 4.3 Out-of-distribution evaluation

In contrast to in-distribution evaluation, we use the models trained with AMR 2.0 data to collect AMR predictions for the testing datasets in the domains that differ from the AMR 2.0 dataset. The purpose of the experiment is to evaluate the ensemble approach under out-of-distribution settings.

---

[6] https://github.com/mdtux89/amr-evaluation

Table 2: Results on the test splits of the AMR 2.0 and AMR 3.0 dataset.

| Models | Smatch | Unl. | NWSD | Con. | NE | Neg. | Wiki. | Reen. | SRL |
|--------|--------|------|------|------|-----|------|-------|-------|-----|
| **AMR 2.0** | | | | | | | | | |
| SPRING | 84.22 | 87.38 | 84.72 | 89.98 | 90.77 | 72.65 | 82.76 | 74.30 | 82.89 |
| APT | 82.70 | 86.18 | 83.23 | 89.48 | 90.20 | 67.27 | 78.87 | 73.19 | 82.01 |
| T5 | 82.98 | 86.17 | 83.43 | 89.85 | 90.65 | 73.43 | 77.99 | 72.44 | 82.02 |
| Cai&Lam | 80.15 | 83.60 | 80.66 | 87.39 | 82.25 | **78.09** | **85.36** | 66.46 | 77.35 |
| *Graphene 4S* | 84.78 | 87.96 | 85.29 | 90.64 | 92.19 | 75.22 | 83.88 | 71.42 | 83.46 |
| *Graphene All* | **85.85** | **88.68** | **86.35** | **91.23** | **92.30** | 77.01 | 84.63 | **74.49** | **84.41** |
| **AMR 3.0** | | | | | | | | | |
| SPRING | 83.25 | 86.40 | 83.71 | 89.38 | 87.80 | 72.94 | 81.22 | 73.33 | 81.97 |
| APT | 80.57 | 83.96 | 81.07 | 88.38 | 86.82 | 68.69 | 76.88 | 70.78 | 80.17 |
| T5 | 82.17 | 85.22 | 82.66 | 89.03 | 86.99 | 72.59 | 73.78 | **72.18** | 81.18 |
| *Graphene 4S* | 83.77 | 86.89 | 84.23 | 90.09 | 88.27 | 74.60 | 81.92 | 70.22 | 82.46 |
| *Graphene All* | **84.41** | **87.35** | **84.83** | **90.51** | **88.64** | **74.76** | **82.25** | 71.93 | **83.15** |

Table 3: Results of out-of-distribution evaluation on the BIO, New3, and Little Prince dataset.

| Models | Smatch | Unl. | NWSD | Con. | NE | Neg. | Wiki. | Reen. | SRL |
|--------|--------|------|------|------|-----|------|-------|-------|-----|
| **BIO** | | | | | | | | | |
| SPRING | 60.52 | 65.33 | 61.42 | 67.76 | **33.92** | 65.68 | 3.80 | 51.19 | 62.86 |
| APT | 51.23 | 56.27 | 51.81 | 58.22 | 15.68 | 52.91 | 3.62 | 43.53 | 54.24 |
| T5 | 58.89 | 63.86 | 59.69 | 66.63 | 30.42 | 65.11 | 2.46 | 48.56 | 61.47 |
| Cai&Lam | 42.22 | 49.78 | 42.85 | 47.10 | 5.19 | 51.42 | **7.32** | 39.23 | 51.00 |
| *Graphene 4S* | 61.51 | 66.22 | 62.28 | 68.48 | 33.02 | 68.24 | 4.46 | 50.40 | 63.70 |
| *Graphene All* | **62.29** | **66.89** | **63.07** | **68.64** | 32.62 | **69.48** | 4.54 | **52.06** | **64.21** |
| **New3** | | | | | | | | | |
| SPRING | 74.66 | 78.99 | 75.21 | 82.38 | 67.52 | 67.48 | 67.20 | 66.47 | 75.65 |
| APT | 71.06 | 75.92 | 71.58 | 80.34 | 65.65 | 67.08 | 57.14 | 63.02 | 73.40 |
| T5 | 73.04 | 77.30 | 73.68 | 82.65 | 68.24 | 64.20 | 56.42 | 64.65 | 75.03 |
| Cai&Lam | 60.81 | 66.00 | 61.29 | 72.79 | 45.60 | 59.57 | 46.39 | 57.70 | 68.87 |
| *Graphene 4S* | 74.84 | 79.23 | 75.30 | 82.56 | **69.98** | 69.51 | **68.34** | 63.53 | 76.31 |
| *Graphene All* | **75.60** | **79.64** | **76.14** | **83.08** | 68.40 | **69.62** | 67.98 | **67.16** | **76.88** |
| **Little Prince** | | | | | | | | | |
| SPRING | 77.85 | 82.31 | 78.85 | 84.68 | 60.53 | 70.72 | 60.53 | **68.28** | 77.78 |
| APT | 75.21 | 80.07 | 76.12 | 85.29 | 65.15 | 67.92 | 69.70 | 63.28 | 75.31 |
| T5 | 77.66 | 81.99 | 78.53 | 85.12 | 58.06 | 72.33 | 59.35 | 67.03 | 78.30 |
| Cai&Lam | 71.03 | 75.91 | 72.07 | 80.18 | 22.73 | 57.51 | 31.50 | 59.29 | 72.02 |
| *Graphene 4S* | 77.91 | 82.40 | 78.86 | 84.91 | 61.54 | 73.58 | 60.65 | 64.77 | 78.12 |
| *Graphene All* | **78.54** | **82.81** | **79.44** | **85.52** | **64.05** | **75.11** | **63.45** | 67.83 | **78.72** |

Table 3 shows the results of our experiments. Similar to the in-distribution experiments, the *Graphene 4S* algorithm achieves better results than other individual models, while the *Graphene All* approach improves the given results further. We achieve the new state-of-the-art results in these benchmark datasets (under out-of-distribution settings). This result has an important practical implication because in practice it is very common not to have labeled AMR data for domain-specific texts. After all, the labeling task is very time-demanding. Using the proposed ensemble methods we can achieve better results with domain-specific data not included in the training sets.

## 4.4 How the ensemble algorithm works

We explore a few examples to demonstrate the reason why the ensemble method works. Figure 3 shows a sentence with a gold AMR in Penman format and a list of AMRs corresponding to the prediction of SPRING [Bevilacqua et al., 2021], T5 [Raffel et al., 2020], APT [Zhou et al., 2021], Cai and Lam [Cai and Lam, 2020b] parser and the ensemble graph given by Graphene.

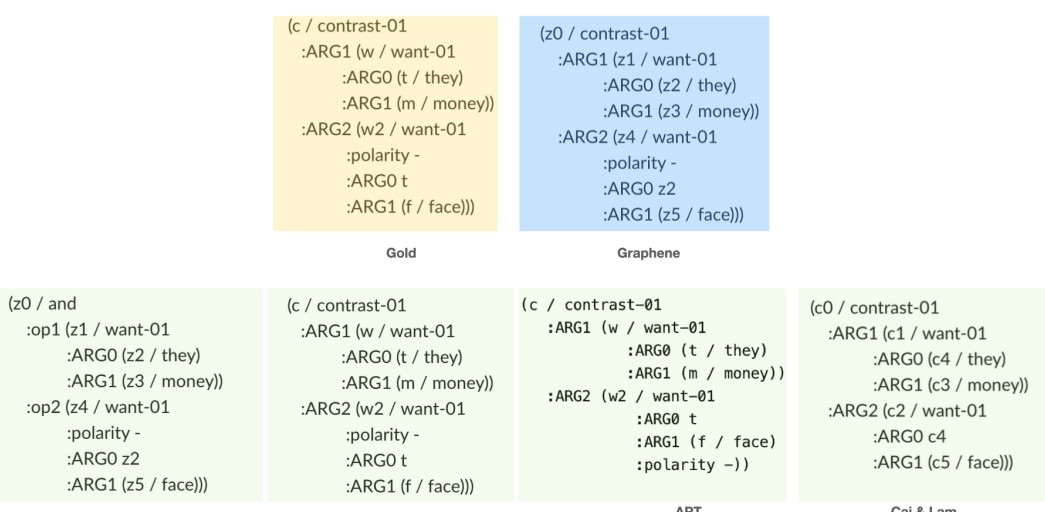

Figure 3: The gold AMR and the ensemble AMR graph of SPRING, T5, APT and Cai&Lam using the Graphene algorithm for the sentence *"They want money, not the face"*.

In this particular example, with the sentence *"They want money, not the face"*, the AMR prediction from SPRING is inaccurate. Graphene corrects the prediction thanks to the votes given from the other models. In particular, the label $and$ of the root node $z_0$ of SPRING prediction was corrected to $contrast - 01$ because T5, APT and Cai&Lam parsers all vote for $contrast - 01$. On the other hand, the labels : $op1$ and : $op2$ of the edges $(z_0, z_1)$ and $(z_0, z_4)$ were modified to have the correct labels : $ARG1$ and : $ARG2$ accordingly thanks to the votes from the other models. We can also see that even though the Cai&Lam method misses polarity prediction, since the other models predict polarity correctly, the ensemble prediction does not inherit this mistake. Putting everything together, the prediction from Graphene perfectly matches with the gold AMR graph in this example.

Table 4: The average total support and Smatch score of SPRING, Graphene with SPRING as a pivot and Graphene respectively. The support is highly correlated with Smatch score.

|  | AMR 2.0 | | AMR 3.0 | | BIO | | LP | | New3 | |
|---|---|---|---|---|---|---|---|---|---|---|
|  | Sup. | Smat. | Sup. | Smat. | Sup. | Smat. | Sup. | Smat. | Sup. | Smat. |
| SPRING | 170.15 | 84.08 | 136.90 | 83.14 | 166.86 | 60.52 | 69.33 | 77.85 | 118.27 | 74.66 |
| SPR. pivot | 172.70 | 84.70 | 139.42 | 83.73 | 169.97 | 61.56 | 70.97 | 78.22 | 120.85 | 74.83 |
| Graphene | **175.73** | **85.85** | **142.07** | **84.43** | **179.38** | **62.29** | **72.64** | **78.54** | **123.62** | **75.60** |

The Graphene algorithm searches for the graph that has the largest support from all individual graphs. One question that arises from this is whether the support is correlated with the accuracy of AMR parsing. Table 4 shows the support and the Smatch score of three models in the standard benchmark datasets. The first model is SPRING, while the second one denoted as *SPR. pivot* uses SPRING prediction as a pivot. The last model corresponds to the Graphene algorithm. Since Graphene looks for the best pivot to have better-supported ensemble graphs, the total supports of the Graphene predictions are larger than the SPR. pivot predictions. From the table, we can also see that the total support is highly correlated to the Smatch score. Namely, Graphene has higher support in all the benchmark datasets and a higher Smatch score than SPR. pivot. This experiment suggests that by optimizing the total support we can obtain the ensemble graphs with higher Smatch score.

## 5   Related work

**Ensemble learning.**   Ensemble learning is a popular machine learning approach that combines predictions from different learners to make a more robust and more accurate prediction. Many ensembling approaches have been proposed, such as bagging [Breiman, 1996] or boosting [Schapire and Freund, 2013], the winning solutions in many machine learning competitions [Chen and Guestrin, 2016]. These methods are proposed mainly for regression or classification problems. Recently,

structure prediction emerges as an important research problem, it is important to study ensemble methods for combining structure predictions.

**Ensemble structure prediction.** Previous studies have explored various ensemble learning approaches for dependency and constituent parsing: [Sagae and Lavie, 2006] proposes a reparsing framework that takes the output from different parsers and maximizes the number of votes for a well-formed dependency or constituent structure; [Kuncoro et al., 2016] uses minimum Bayes risk inference to build a consensus dependency parser from an ensemble of independently trained greedy LSTM transition-based parsers with different random initializations. Note that a syntactic tree is a special graph structure in which nodes for a sentence from different parsers are roughly the same. In contrast, we propose an approach to ensemble graph predictions in which both graph nodes and edges can be different among base predictions.

**Ensemble methods for AMR parsing.** Parsing text to AMR is an important research problem. State-of-the-art approaches in AMR parsing are divided into three categories. Sequence to sequence models [Bevilacqua et al., 2021, Konstas et al., 2017, Van Noord and Bos, 2017, Xu et al., 2020] consider the AMR parsing as a machine translation problem that translates texts to AMR graphs. The transition-based methods [Zhou et al., 2021] predicts a sequence of actions given the input text, and then the action sequence is turned into an AMR graph using an oracle decoder. Lastly, graph-based methods [Cai and Lam, 2020b] directly construct the AMR graphs from textual data. All these methods are complementary to each other and thus ensemble methods can leverage the strength of these methods to create a better prediction, as demonstrated in this paper. Ensemble of AMR predictions from a single type of model is studied in [Zhou et al., 2021], by combining predictions from three different model's checkpoints it gains performance improvement in the final prediction. However, ensemble in sequential decoding requires that all predictions are from the same type of models. It is not applicable for cases when the predictions are from different types of models such as seq2seq, transition-based or graph-based models. In contrast to that approach, our algorithm is model-agnostic, i.e. it can combine predictions from different models. In our experiments, we have demonstrated the benefit of combining predictions from different models, with additional gains in performance compared to the ensemble of predictions from a single model's checkpoints.

**Comparison to Bazdins et al.** [7] Barzdins and Gosko [2016] proposed a character-level based neural method for parsing texts into AMRs. To improve the robustness of the parser, an ensemble technique which selects among the prediction graphs the one that has the highest average SMATCH when it is compared against the other predictions was proposed. The key difference between Barzdins' and our approach is that while our solution modifies the predictions to create new prediction candidates for ensemble prediction, Barzdins' approach only selects a prediction among existing predictions. For a detailed discussion with additional experimental results please refer to Lam et al. [2021]

## 6 Conclusions and future work

In this paper, we formulate the graph ensemble problem, study its computational intractability, and provide an algorithm for constructing graph ensemble predictions. We validate our approach with AMR parsing problems. The experimental results show that the proposed approach outperforms the previous state-of-the-art AMR parsers and achieves new state-of-the-art results in five different benchmark datasets. We demonstrate that the proposed ensemble algorithm not only works well for in-distribution but also for out-of-distribution evaluations. This result has a significant practical impact, especially when applying the proposed method to domain-specific texts where training data is not available. Moreover, our approach is model-agnostic, which means that it can be used to combine predictions from different models without the requirement of having an access to model training. In general, our approach provides the basis for graph ensemble, studying classical ensemble techniques such as bagging, boosting, or stacking for graph ensemble is a promising future research direction that is worth considering to improve the results further.

---

[7]We would like to thank Juri Opitz for pointing us to missing references as a very useful feedback for our published work.

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
