# Ensemble Graph Prediction
# Supplementary Material

## A  Running time

Figure 4 shows the average running time of the Graphene algorithm. The horizontal axis corresponds to the average graph size (the number of triples) while the vertical axis shows the average running time (in seconds). We can see that the running time depends on the average size of the AMR graphs. Since AMR graph size is proportional to the input sentence length, the largest average graph has around 50 triples. Graphene requires less than 2 seconds on an 8-core CPU machine to make an ensemble from 7 models.

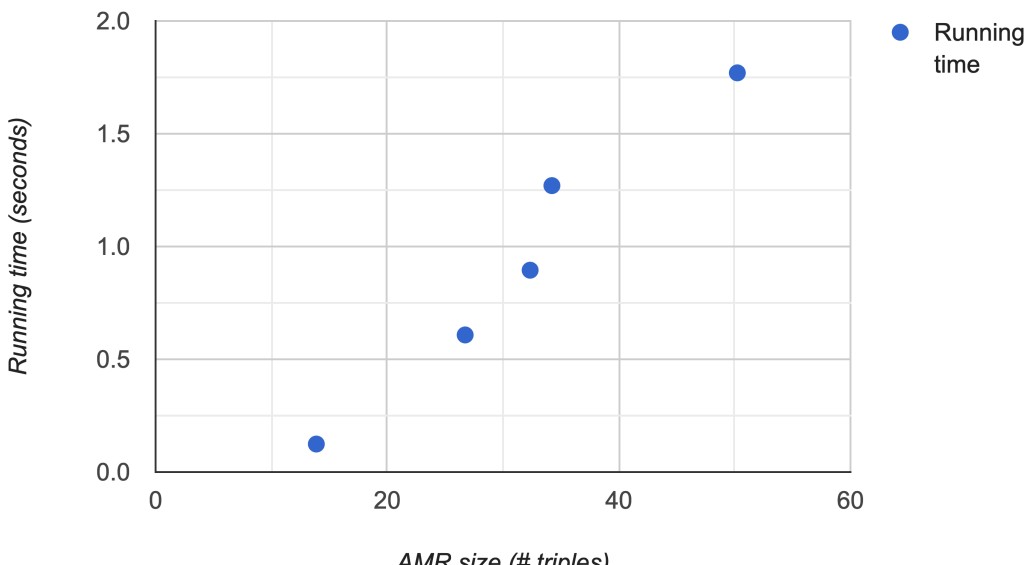

Figure 4: Average running time of the Graphene algorithm versus the average graph size in the LP, New3, AMR 3.0, AMR 2.0, and BIO datasets respectively.

## B  On the support threshold

The popular VotingClassifier algorithm implemented in scikit-learn [8] follows the majority vote rule where the label with the most votes is selected as the final prediction. Therefore, we apply the same rule in our experimental settings where setting theta = 0.5 is equivalent to the majority vote rule in classification problems.

If there is an independent validation set, this hyper-parameter can be tuned to choose the right theta value for that dataset. For example, in the AMR 2.0 dataset, the results of ensembling 4 Spring

---

[8]https://scikit-learn.org/stable/modules/generated/sklearn.ensemble.VotingClassifier.html

models, APT model, and T5 models on the validation set (the dev split) when theta is varied are reported in Table 5.

Table 5: The results of ensembling 4 Spring models, APT model, and T5 models on the validation set (the dev split) when $\theta$ is varied. On this dev set, $\theta = 0.5$ is a proper choice for AMR 2.0.

|  | $\theta = 0.1$ | $\theta = 0.3$ | $\theta = 0.5$ | $\theta = 0.7$ | $\theta = 0.9$ |
|---|---|---|---|---|---|
| Smatch | 81.64 | 85.01 | **85.49** | 85.12 | 83.68 |
| Precision | 76.13 | 83.62 | 85.86 | 86.53 | **86.54** |
| Recall | **88.00** | 86.55 | 85.13 | 83.75 | 81.01 |

Based on this result on an independent dev set, theta=0.5 is the right choice for AMR 2.0. Note that setting theta is a trade-off between precision and recall.

## C Comparison with median baselines

Beside the Graphene 4S baseline, we provide the results in Table 6 of the following baseline approaches:

- Uniform sampling: for each set of predictions we sample the graph uniformly at random, this approach is equivalent to the "median" representative from a set.

- Ideal median: assumes that the gold AMRs are available for the test set (hence named as "ideal"). We computed the Smatch of each prediction with the gold AMR and use the AMR with the median Smatch score as the final prediction.

Table 6: Comparison with median baselines

|  | **AMR 2.0** | **AMR 3.0** | **BIO** | **new3** | **LP** |
|---|---|---|---|---|---|
| uniform Sampling | 82.58 | 82.98 | 56.00 | 71.18 | 76.17 |
| Ideal median | 83.80 | 83.66 | 57.72 | 73.06 | 77.14 |
| Graphene | **85.85** | **84.41** | **62.29** | **75.60** | **78.54** |

## D Pivot selection

Figure 5 shows the pie-charts with the percentage summarizing the number of times that the prediction created when each algorithm is chosen as a pivot graph is selected as the final prediction. Notice that the order of the algorithms matters because when tight happens, the ensemble is chosen from the first algorithm in the list.

The results show that all algorithms contribute to the final predictions. In the Bio dataset where the test data is from a specific domain that differs from the training data domain, Graphene benefits from the model diversity when it leverages predictions from all models effectively.

## E Robustness on down-sampled training data

Table 7: When the training data is down-sampled, the gain of using Graphene is enlarged.

| Methods/Data | Sample rate 0.6 | | | | Sample rate 0.8 | | | |
|---|---|---|---|---|---|---|---|---|
|  | AMR 2.0 | BIO | New3 | LP | AMR 2.0 | BIO | New3 | LP |
| SPRING 603 | 82.40 | 58.22 | 73.81 | 76.75 | 83.28 | 58.85 | 74.13 | 76.96 |
| SPRING 703 | 82.74 | 57.93 | 73.49 | 76.60 | 83.43 | 59.50 | 74.71 | 77.40 |
| SPRING 803 | 82.85 | 58.72 | 73.42 | 76.68 | 83.39 | 58.97 | 73.57 | 77.39 |
| SPRING 903 | 82.81 | 58.80 | 74.09 | 76.96 | 83.12 | 57.52 | 73.50 | 76.70 |
| T5 | 82.08 | 56.46 | 72.84 | 76.31 | 82.59 | 58.69 | 73.42 | 77.70 |
| Graphene | **84.20** | **61.66** | **75.01** | **77.79** | **84.70** | **62.23** | **75.98** | **78.09** |

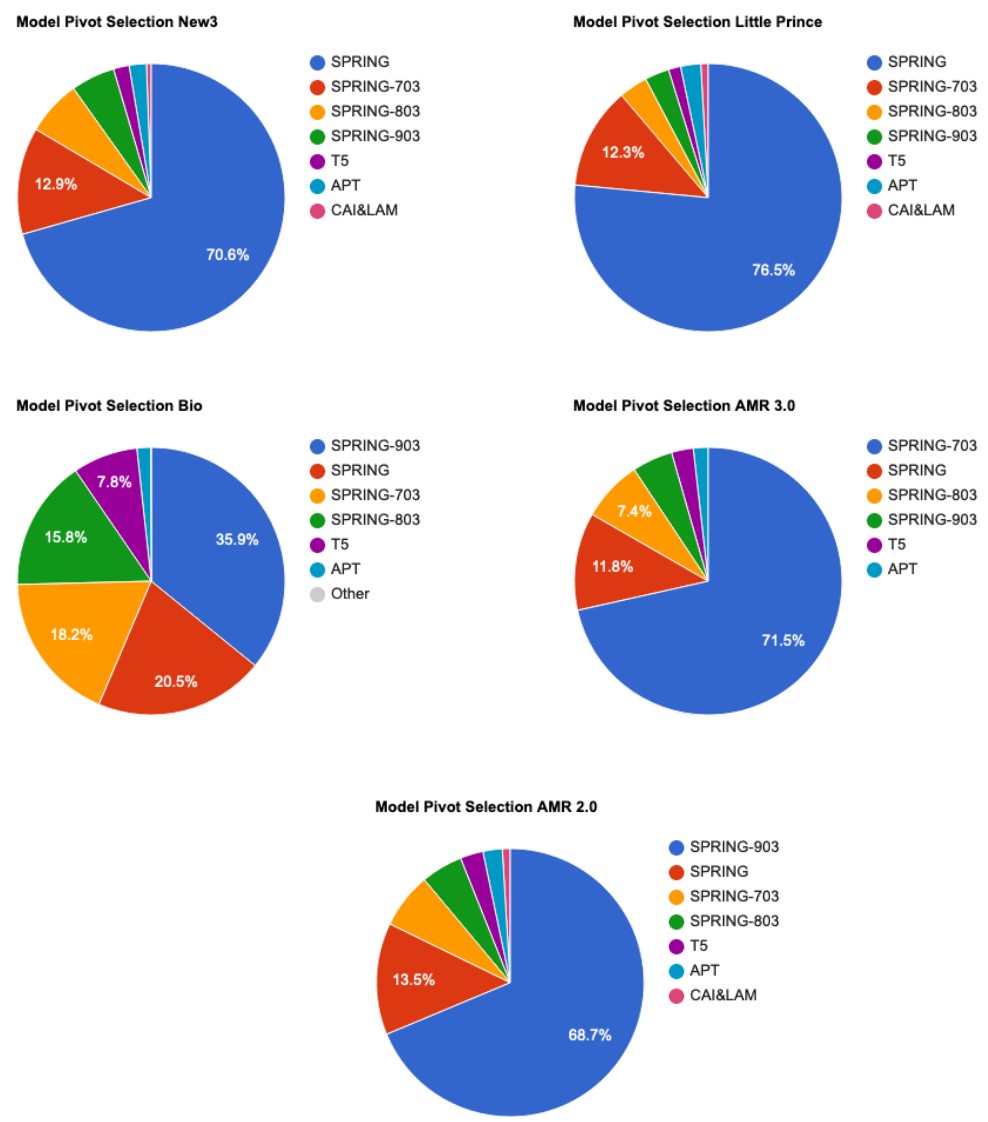

Figure 5: Pie charts shows the percentage of the number of times each model was selected as the best pivot in the Graphene algorithm. Notice that when tight happens, the ensemble created when first algorithm is considered as the pivot is selected as the final ensemble graph.

We down-sampled the AMR 2.0 training data with sample rates 0.6 and 0.8. Then 4 Spring models with different random seeds and the T5 model were trained on these two sample sets. The Smatch score on AMR 2 test sets and on the out-of-distribution sets (LP, New3, Bio) are reported in Table 7.

Compared to the best individual models, Graphene is more robust and 1.35, 2.86, 0.92, and 0.83 points better when the sample rate is equal to 0.6. While compared to the best individual models, Graphene is more robust and 1.27, 2.73, 1.27 and 0.39 points better better when the sample rate is equal to 0.8. This result demonstrates that the proposed method is robust with respect to smaller training data.

## F   Ties broken arbitrarily

When many ensemble graphs have the same support, Graphene chooses the ensemble graph created when the first model in the list is chosen as the pivot. Table 8 shows the results when each model is put first in the list. If we have a validation set like the case with AMR 2.0 or AMR 3.0, we can tune the right input order to achieve the best performance on the validation set.

In case there is no validation set available, to mitigate the impact of random input order, we can break the ties arbitrarily, the results of ties broken arbitrarily are reported in Table 9.

Table 8:  Results of Graphene when each model is put first in the list.

| Grapphene Models | S | S703 | S803 | S903 | T5 | APT | Cai&Lam |
|---|---|---|---|---|---|---|---|
| AMR 2.0 | 85.66 | 85.77 | 85.81 | **85.87** | 85.65 | 85.67 | 85.11 |
| AMR 3.0 | **84.44** | 84.42 | 84.34 | **84.44** | 84.35 | 84.18 | NA |
| Bio | 62.38 | 62.41 | 62.39 | **62.44** | 62.38 | 62.41 | 62.34 |
| LP | 78.65 | 78.63 | **78.75** | 78.70 | 78.65 | 78.23 | 77.75 |
| New3 | 75.65 | 75.69 | **75.88** | 75.82 | 75.78 | 75.48 | 74.92 |

Table 9:  Results of Graphene when ties are broken arbitrarily.

| Grapphene Models | AMR 2.0 | AMR 3.0 | Bio | LP | New3 |
|---|---|---|---|---|---|
| Graphene 4SATC | 85.52 | 84.27 | 62.39 | 78.50 | 75.66 |

## G   Support and Smatch

We have shown in Table 4 that the average total support is highly correlated with the Smatch score. We performed statistical significant tests to support the given hypothesis. Below is the correlation between the "Normalised total support" (the total support normalised to the size of the graph) and the Smatch score, together with the p-value for each dataset:

- AMR 2.0: Pearson correlation =0.60 , p-value = 2.7e-117
- AMR 3.0: Pearson correlation =0.49 , p-value = 2.6e-137
- BIO: Pearson correlation =0.55 , p-value = 0.0
- LP: Pearson correlation =0.56, p-value =3.4e-130
- New3: Pearson correlation = 0.73, p-value=5.1e-191

The overall correlation between the "Normalised total support" and the Smatch score, together with the p-value for all datasets is: Pearson correlation =0.67 , p-value=0.0