# OpenReview forum: "Ensembling Graph Predictions for AMR Parsing"
_NeurIPS.cc/2021/Conference — NeurIPS 2021 Poster_

### Official Review · Reviewer_awK5 · 2021-07-13

**Rating:** 6
**Confidence:** 5

**Summary:**

The authors present a heuristic to combine several graphs, obtained by specialised linguistic parsers, to obtain a single more accurate parse graph.

**Limitations And Societal Impact:**

The authors have not addressed the limitations and potential negative societal impact of their work.

**Main Review:**

The main concern with this work lies in the benchmarking design. Specifically:
1. Size of the improvement:  the authors claim that since their proposal is deterministic they do not need to report the variance in the experimental results, however one should simply sample the training set to observe the variability of the proposed approach. All improvements reported in the result section are around the 1% mark, and is therefore difficult to exclude a random effect.
2. No base line is offered: for example one could propose the "median" graph as a representative of the set, or one could select the best graph by formulating a re-ranking task (i.e. learn a preference score on the set of alternatives). The efficacy of the approach (i.e. deriving an adjusted graph from a set of graphs) is not compared against any approach or baseline that has in input a set of graphs and outputs a single graph.
3. The algorithm (which is heavily based on prior work to solve the vertex matching problem) does not offer any guarantee of being able to preserve some characteristics of the graph set: for example if one is working on trees, the algorithm is not guaranteed to output a tree, if one was working on molecular graphs, the algorithm would not yield valid molecular graphs (the valence would not be guaranteed). However no mention of the necessity to check or compensate for the feasibility  of the resulting graph is mentioned in this work. It is not clear in this specific domain what characteristics of the graphs are responsible for the efficiency of the algorithm, it is also not clear if this approach could be applied effectively to other domains and hence be of broader interest for the community.
4. Not self contained: the authors do not provide any description of what an Abstract Meaning Representation graph is (apart from a single example figure); the datasets are not described; even the evaluation metric uses terms that are domain specific and not introduced.

###After the authors' reply
Many thanks to the authors for the their response. Given the new experimental data provided by the authors I am inclined to improve the score. While the issues at points 1 and 2 regarding the effect size of the improvement achieved by their proposal and the baseline comparisons are somewhat addressed by the additional experiments, the concern regarding the generality of the method is not, as it is not sufficient to say that one can always introduce constraints to select the the non-violating examples (in fact in many important cases it can become exponentially hard to find viable candidates and hence a generate and test strategy would became infeasible). The generality problem could be easily addressed by specifying that the proposal is tailored for AMR graphs rather than for general graphs (given also that all experiments are in this very domain).

**Time Spent Reviewing:**

2

---

> ### Author Response · Authors · 2021-08-10
> **Additional baseline comparison, robust results on sampled training data, size of improvement of SOTA, constraints on validity of output graphs**
>
>
> First of all, we would like to thank you very much for your useful suggestions and constructive reviews!
>
>
>
> **Regarding the comment**: "*Size of the improvement*"
>
>
>
> **Answer**: for context, we show SoTA Smatch scores from 2018 onwards
>
> -   2018 "AMR parsing as graph prediction with latent alignment" 74.4
> -   2019 "Broad-coverage semantic parsing as transduction" 77.0
> -   2020 "AMR parsing via graph-sequence iterative inference" 80.2
> -   2021 "One SPRING to rule them both: Symmetric amr semantic parsing and generation without a complex pipeline" 84.3
> -   (anon) "Structure-aware Fine-tuning of Sequence-to-sequence Transformers for Transition-based AMR Parsing" 84.7
>
> as it can be seen, performance has experimented a very large increase. When reported, standard deviations in these models are around 0.1-0.2. In this context we consider the ensembling strategy to provide significant improvements.
>
>
>
> **Regarding the comment**: *"the authors claim that since their proposal is deterministic they do not need to report the variance in the experimental results, however one should simply sample the training set to observe the variability of the proposed approach. All improvements reported in the result section are around the 1% mark, and is therefore difficult to exclude a random effect."*
>
>
> **Answer**: we follow the same standard benchmarking settings [Cai&Lam, Bevilacqua et al. and Zhou et al.] with predefined train/dev/test splits for AMR 2.0, AMR 3.0, and out-of-distribution tests for New3, LP, and BIO. Using a standard benchmarking setting enables future work to compare directly to our work. Below are additional experiment results following your suggestions. In particular, we down-sampled the AMR 2.0 training data with sample rates 0.6 and 0.8. Then 4 Spring models with different random seeds and the T5 model were trained on these two sample sets. The Smatch score on AMR 2 test sets and on the out-of-distribution sets (LP, New3, Bio) are reported as follows:
>
>
>
> Sampled AMR 2.0 training data, sample rate = 0.6
>
> |  | AMR 2.0 | BIO | New3 | LP
> |--|--|--|--|--|
> |SPRING 603 | 0.823 | 0.582 | 0.738 | 0.785
> |SPRING 703 | 0.827 | 0.579 | 0.734 | 0.783
> |SPRING 803 | 0.828 | 0.586 | 0.734 | 0.784
> |SPRING 903 | 0.827 | 0.587 | 0.741 | 0.787
> |T5 | 0.819 | 0.565 | 0.728 | 0.781
> |Graphene | **0.843** | **0.617** | **0.750** | **0.796**
>
>
> Compared to the best individual models, Graphene is more robust and 1.5, 3.0, 1.2, and 0.9 points better.
>
>
> Sampled AMR 2.0 training data, sample rate = 0.8
>
>
> |  | AMR 2.0 | BIO | New3 | LP
> |--|--|--|--|--|
> |SPRING 603 | 0.832 | 0.588 | 0.741 | 0.787
> |SPRING 703 | 0.834 | 0.595 | 0.746 | 0.791
> |SPRING 803 | 0.833 | 0.589 | 0.735 | 0.791
> |SPRING 903 | 0.831 | 0.575 | 0.734 | 0.785
> |T5 | 0.824 | 0.586 | 0.734 | 0.795
> |Graphene | **0.848** | **0.622** | **0.760** | **0.799**
>
>
> Compared to the best individual models, Graphene is more robust and 1.4, 2.7 1.4 and 0.4 points better.
>
>
> **Regard the comment**: *"No base line is offered: for example one could propose the "median" graph as a representative of the set, or one could select the best graph by formulating a re-ranking task (i.e. learn a preference score on the set of alternatives). The efficacy of the approach (i.e. deriving an adjusted graph from a set of graphs) is not compared against any approach or baseline that has in input a set of graphs and outputs a single graph."*
>
>
>
> **Answer:** an important point to note here is that this work is the first work that studies ensemble methods for combining AMR graphs instead of sequences. Before applying advanced ensembling algorithms like bagging (bootstrapping), weighted voting, etc. we need a technique that allows us to combine graph predictions. We focus on the development of this basic technique in this paper as a starting point for the community to further apply existing learning ensemble strategies. We leave this research direction opened as a future work as discussed in the conclusion section of the paper.
>
> In the paper, we have compared to a baseline denoted as Graphene 4S when 4 Spring models (at different checkpoints) are combined. Table 2 and 3 show that combining predictions from a diverse set of models yield better results than combining predictions from a single type of model with different checkpoints.
>
> At test time we need a method to rank the ensembled pivot graphs. The metrics that we proposed in the paper are based on the supports counted from all predictions. It is an unsupervised ranking approach that does not require any additional labeled training data.
> On the other hand, in order to learn a preference score on the set of alternatives using supervised learning we need additional labeled training data to train a good classifier to predict model performance given an input prediction. Acquisition of additional training data is not a reasonable task, especially in out-of-domain settings. Moreover, the classifiers should be reasonably accurate so that it is able to predict a good representative from the set of predictions. Whether we can predict the model's performance given an input prediction is an open research question that needs to investigate more in the future.
>
> Based on your suggestions we have implemented and compared the following baselines:
>
> -   **APT checkpoints ensemble**: In Table 2 of the paper "AMR Parsing with Action-Pointer Transformer", which we denote as APT, the authors report the scores from the ensemble that uses the average probability of 3 models from different seeds. The performance improvement ranges from 0.6 to 1.0 smatch point. When the baseline model performance is lower, i.e. 81.8, the improvement is 1.0. When the baseline model performance is higher, i.e. 82.8 (using 70k silver data), the improvement is 0.6 on AMR2.0. Our in-house experiments also indicate that as the baseline parser improves beyond 83.0, the impact of the probabilistic ensemble tends to decrease, consistent with the results reported in the OpenReview-anonymous paper "Structure-aware Fine-tuning of Sequence-to-sequence Transformers for Transition-based AMR Parsing", where the authors report just 0.2 smatch improvement on AMR2.0 and 0.4 smatch improvement on AMR3.0 with the probabilistic ensemble technique in Table 1. On the other hand, as shown in Tables 2 and 3, our proposed technique achieves the performance gains of 1.3 -1.5 smatch points on the standard benchmark data sets and 0.9 - 1.8 on the out-of-distribution data sets, despite the fact that the highest performing system score is 84.1 on AMR2.0 and 83.1 on AMR3.0. Consistent high-performance gains across various datasets combined with the applicability to a diverse set of system outputs indicate the superiority of our proposed ensemble technique over other readily available model-specific ensemble methods.
> -   **Uniform sampling**: for each set of predictions we sample the graph uniformly at random, this approach is equivalent the the "median" representative from a set
> -   **Ideal median**: assumes that the gold amrs are available for the test set (hence named as "ideal"). We computed the SMATCH of each prediction with the gold amr and use the amr with the median SMATCH score as the final prediction.
>
> |  | AMR 2.0 | AMR 3.0 | BIO | New3 | LP
> |--|--|--|--|--|--|
> |Uniform sampling | 0.824 | 0.828 | 0.560 | 0.711 | 0.779
> |Ideal median | 0.835 | 0.836 | 0.577 | 0.731 | 0.788
> |Graphene | **0.858** | **0.844** | **0.622** | **0.757** | **0.804**
>
>   **Regarding the comment:** *"The algorithm (which is heavily based on prior work to solve the vertex matching problem) does not offer any guarantee of being able to preserve some characteristics of the graph set: for example if one is working on trees, the algorithm is not guaranteed to output a tree, if one was working on molecular graphs, the algorithm would not yield valid molecular graphs (the valence would not be guaranteed). However no mention of the necessity to check or compensate for the feasibility of the resulting graph is mentioned in this work. It is not clear in this specific domain what characteristics of the graphs are responsible for the efficiency of the algorithm, it is also not clear if this approach could be applied effectively to other domains and hence be of broader interest for the community."*
>
>
>
> **Answer:** In our paper, page 4 (line 129 to line 135) we have discussed a solution to guarantee the validity of the output graphs, for example:
>
> *"For special cases, when disconnected graphs are not considered as a valid output, we keep all edges of the pivot graph even its support is below the threshold.  On the other hand, for the graph prediction problem, where a graph is only considered a valid graph if it does not have multiple edges between two vertices and multiple labels for any vertex,  we remove all candidate labels for vertices and edges  except the one with the highest number of votes."*
>
> We applied a set of constraints to avoid generating invalid graphs. For different domains with different types of graphs, the set of constraints might vary but for AMR graphs, connectivity, node and edge label constraints help guaranteeing the correctness of the output AMR.
>
>
>
> **Regarding the comment:** *"Not self contained: the authors do not provide any description of what an Abstract Meaning Representation graph is (apart from a single example figure); the datasets are not described; even the evaluation metric uses terms that are domain specific and not introduced."*
>
>
>
> **Answer:** Thank you we will provide an additional paragraph at the beginning of the paper to briefly introduce the basic concept of AMR graph and a description of the datasets to make it more clear.

---

> > ### Author Response · Authors · 2021-08-25
> > **Feedback on response**
> >
> > Dear anonymous reviewer,
> > As the deadline for the review discussion period is approaching, could you please kindly provide us with some feedback on our response to your comments?
> > Any comment is highly appreciated!
> > Thank you very much and best regards,

---

### Official Review · Reviewer_kDjg · 2021-07-14

**Rating:** 7
**Confidence:** 4

**Summary:**

The authors proposed a new method for ensemble graph prediction. They formalized this problem as "mining the largest subgraph that is the most supported by a collection of graph predictions". Furthermore, they introduced an efficient heuristic algorithm to approximate the optimal solution given that the original problem is NP-Hard. They tested the algorithm in the AMR parsing problems. The experimental results demonstrate that the proposed approach can combine the strength of the state-of-the-art AMR parsers and the ensemble model outperforms individual models in five standard benchmark datasets.

**Limitations And Societal Impact:**

There's no foreseeable negative societal impact of this work.

**Main Review:**

The work is innovative as previous work focuses on dealing with multiple predictions where nodes are the same but the current work deals with the situation where both graph nodes and edges can be different among base predictions. Furthermore, previous ensemble approaches require models to be of the same type such as sequence models, transition-based or graph-based models. But the current approach is model-agnostic in the sense that it can combine predictions from different models.

 The submission is relatively sound. The authors demonstrate that their ensemble approach outperforms individual models not only in the In-distribution setting but also in the Out-of-distribution setting. I do have two concerns. First, in section 4.4, the authors claim that "From Table(4), the total support is highly correlated to the Smatch score. " Without seeing the statistical test of the correlation coefficients, I'm not entirely convinced that the conclusion is true. Even in Table 4, there're cases in which the average support is high but the Smatch score is low (e.g., BIO in row 3). It would be nice if the author could report correlation coefficients and the p-values to support their claim. Also, the authors mention that they set the hyperparameter theta so that theta divided by the number of models is greater than 0.5 (Line 180). I was wondering how changing this hyperparameter would affect the results and whether tuning this hyperparameter would lead to any further improvement.

The submission is clearly written and well organized. The results are important in the sense that it provides a way to obtain the ensemble prediction from multiple models, even when the models being combined have different architectures. I was wondering if the authors have any intuitions on how to generalize their work to graphs more complicated than the AMR parsing trees. It seems that their heuristic algorithms only work for a subset of graphs. Therefore naming the paper as "ensemble graph prediction" seems to be a bit overselling.

---------------

Updated: I've read the authors' responses to my review as well as the ones to the other reviewers'. I think their responses address my concerns, especially the additional details on the correlation analysis and the justification for choosing the hyper-parameter theta. I don't have a strong preference for adding "AMR" to the title, since I'd like to keep the paper appealing to a broad audience. I'm looking forward to seeing more work in the future which could verify that this ensemble approach is not limited to AMR graphs but can be applied to other graphs as well. Given the improvements and the limitation of the paper, I decided to keep my score unchanged as it reflects my best judgment of the work.

**Time Spent Reviewing:**

10 hours

---

> ### Author Response · Authors · 2021-08-10
> **Correlation and p-value analysis, hyper-parameter tuning on dev set and majority vote rule.**
>
>
> First of all, we would like to thank you very much for your useful suggestions and constructive reviews!
>
> **Regarding the comment**: *"First, in section 4.4, the authors claim that "From Table(4), the total support is highly correlated to the Smatch score. " Without seeing the statistical test of the correlation coefficients, I'm not entirely convinced that the conclusion is true. Even in Table 4, there're cases in which the average support is high but the Smatch score is low (e.g., BIO in row 3). It would be nice if the author could report correlation coefficients and the p-values to support their claim. "*
>
>
>
> **Answer:** the average total support reported in Table 4 is the average of the sum of the support of all the nodes and the edges in an ensembled graph. Because the number is not normalized to the size of the graph we can see that its magnitude is varied depending on datasets. For instance, in the BIO dataset where the input sentences and the AMR graphs are usually larger, the average sum of support is also larger accordingly. Our claim in section 4.4 concerns the correlation of the average total support and the SMATCH score within each dataset.
>
> In particular, for the BIO dataset (row 3 Table 4) we can see that the total support and corresponding SMATCH are (166.86 60.52), (169.97 61.57), and (179.38 62.27) showing an almost perfect correlation between these two scores. We will add a concrete definition of the "Total support" and change the title of the column in Table 4 from "Avg. support" to "Total support" to make it more clear.
>
> Below is the correlation between the "Normalised total support" (the total support normalised to the size of the graph) and the SMATCH score, together with the p-value for each dataset:
>
> -   AMR 2.0: Pearson correlation =0.628 , p-value = 0.0
> -   AMR 3.0: Pearson correlation =0.505 , p-value = 0.0
> -   BIO: Pearson correlation =0.526 , p-value = 0.0
> -   LP: Pearson correlation =0.547, p-value =0.0
> -   New3: Pearson correlation = 0.725, p-value=2.26e-258
>
> The overall correlation between the "Normalised total support" and the SMATCH score, together with the p-value for all datasets is: Pearson correlation =0.68 , p-value=0.0
>
>
>
> **Regarding the comment**: *"Also, the authors mention that they set the hyperparameter theta so that theta divided by the number of models is greater than 0.5 (Line 180). I was wondering how changing this hyperparameter would affect the results and whether tuning this hyperparameter would lead to any further improvement."*
>
>
>
> **Answer**: the popular VotingClassifier algorithm implemented in scikit-learn (see [https://scikit-learn.org/stable/modules/generated/sklearn.ensemble.VotingClassifier.html](https://scikit-learn.org/stable/modules/generated/sklearn.ensemble.VotingClassifier.html)) follows the majority vote rule where the label with the most votes is selected as the final prediction. Therefore, we apply the same rule in our experimental settings where setting theta = 0.5 is equivalent to the majority vote rule in classification problems.
>
> If there is an independent validation set, this hyper-parameter can be tuned to choose the right theta value for that dataset. For example, in the AMR 2.0 dataset, the results of ensembling 4 Spring models, APT model, and T5 models on the validation set (the dev split) when theta is varied are reported as follows:
>
>  **SMATCH on the independent dev set on AMR 2.0**
>
> || theta=0.1 | theta=0.3 | theta=0.5 | theta=0.7 | theta=0.9
> |--|--|--|--|--|--|
> |SMATCH | 0.806 | 0.854 | **0.857** | 0.848 | 0.831
> |Precision | 0.738 | 0.826 | 0.854 | 0.866 | 0.864
> |Recall | 0.887 | 0.873 | 0.861 | 0.830 | 0.801
>
>
> Based on this result on an independent dev set, theta=0.5 is the right choice for AMR 2.0. Note that setting theta is a trade-off between precision and recall.
>
> **Regarding the comment**: "*The submission is clearly written and well organized. The results are important in the sense that it provides a way to obtain the ensemble prediction from multiple models, even when the models being combined have different architectures. I was wondering if the authors have any intuitions on how to generalize their work to graphs more complicated than the AMR parsing trees. It seems that their heuristic algorithms only work for a subset of graphs.Therefore naming the paper as "ensemble graph prediction" seems to be a bit overselling.*"
>
>
> **Answer**: even the implementation and experiments are mainly demonstrated for AMR graphs, the method that combines graph structure predictions is general and it enables further research on applying advanced ensembling methods like bagging, weighted voting, etc. However, we are happy to revise the paper title to "Graph Ensemble Prediction for AMR Parsing".

---

### Official Review · Reviewer_qSVh · 2021-07-17

**Rating:** 6
**Confidence:** 3

**Summary:**

Ensemble learning is a popular machine learning practice, in which predictions from multiple models are blended to create a new one that is usually more robust and accurate. Lots of work show the benefits of ensemble learning on regression or classification problems but there is a lack of investigation on the problem of graph ensemble which is more complicated. The paper collects the predictions from four state-of-the-art AMR parsers and shows the proposed ensemble method's effectiveness on the standard benchmark datasets.

**Limitations And Societal Impact:**

1. I didn't see too many experiments on model analysis that could reveal the insights of the proposed model.

2. Minor: The title of section 3, 'Please correct me if I am wrong', is quite interesting to me. It reflects the model's behavior and explicitly concludes the proposed method in a straightforward way. However, there is one more meaning that corrects you if you have something wrong in this section. I am not sure if you imply this meaning or not. (Although I really like this title, I suggest you use a conventional title to avoid any ambiguity)

**Main Review:**

1. The motivation of the paper to study ensemble learning over graphs (specifically on the AMR task) is quite clear and the tackling problem is very important and crucial.

2. The proposed method which aims to figure out the largest supported common subgraph makes sense to me.

3. The experimental results of this paper show massive numbers to show the effectiveness of the proposed method in ensembling graphs. The performance improvement from the proposed method is consistent and promising.

**Time Spent Reviewing:**

1

---

> ### Author Response · Authors · 2021-08-10
> **Section title revision and result analysis**
>
>
> First of all, we would like to thank you very much for your useful suggestions and constructive reviews!
>
> **Regarding the comment**: "*I didn't see too many experiments on model analysis that could reveal the insights of the proposed model.*"
>
> **Answer:** We have provided the following additional analyses to give some insights on how the ensemble models work:
>
> -   We did more experiments with sampled training data to show the robustness (see our response to Reviewer 4) and experiments to demonstrate the reason why theta = 0.5 (see our response to Reviewer 3), reported correlation and p-value between normalized support and smatch scores which shows why optimizing support leads to a better smatch score.
>
> Also, we provide the following insights in the paper and in the appendices:
>
> -   Qualitatively, in section 4.4, Figure 3 shows an example where SPRING prediction was wrong but with the help of other models, the prediction was corrected.
> -   In section 4.4 Table 4, we reported the average of the total support of the ensembled graphs. The results explain why optimizing the support helps to improve the SMATCH score.
> -   In the appendix available with the supplementary materials, section 2 Figure 2, we reported Pie charts showing the percentage of the number of times each model was selected as the best pivot in the Graphene algorithm. The results revealed that in most cases the best pivot was selected from the best model but for new domain-specific data such as in the BIO dataset, the best pivot was selected across all 4 types of models.
> -   Running example in Figure 2 also demonstrates the intuition behind the algorithms.
>
>
>
> **Regarding the comment:** *"However, there is one more meaning that corrects you if you have something wrong in this section."*
>
> **Answer:** thank you for a good suggestion we will change the title of the section to "Graph ensemble algorithm" to avoid any ambiguity.

---

> > ### Author Response · Authors · 2021-08-25
> > **Review response discussion**
> >
> > Dear anonymous reviewer,
> >
> > As the deadline for the review discussion period is approaching, could you please kindly provide us with some feedback on our response to your comments?
> >
> > Any comment is highly appreciated!
> >
> > Thank you very much and best regards,

---

### Official Review · Reviewer_7y9L · 2021-07-17

**Rating:** 7
**Confidence:** 3

**Summary:**

The paper presents an ensemble approach for graph predictions.
It consists in an algorithm that finds the graph that have the most nodes and edges matching, among different graphs produced by different systems or different checkpoints of the same system.
The authors tested the system on AMR graphs produced by state of the art AMR parsers.
Even if the problem is NP-hard the authors made it tractable using heuristics

The proposed ensemble models outperform the single-model approaches the authors compare with in two used AMR benchmarks in both in-distribution and out-of-distribution setting.

UPDATE
Thanks for the complete response, I think the paper is much stronger now.

**Limitations And Societal Impact:**

There is no discussion on the limitations of the work and societal impact, but I am not sure it can apply to this paper.

**Main Review:**

The paper is easy to read and the approach is simple (which is good, it only has one hyperparameter) and effective. It will be likely used to boost performances of the next generation of AMR-parsers.

I liked the explanation the authors give on how the algorithm work in Section 4.4.

Since the authors only tested the model on AMR graphs, they should mention it in the title. The approach is clearly general and it would be interesting see its application on other graph prediction tasks.

The main thing missing in this paper in my opinion is an ensemble baseline.
For example, the transition base parser could be easily ensembled.
Another type of ensembling could be stacking in sequence to sequence, where the output of a model is used as input for another model.

I understand that the way the authors proposed to merge graphs is novel, but I would also like to see a comparison with more "standard" ensembling approaches.



**Time Spent Reviewing:**

3

---

> ### Author Response · Authors · 2021-08-10
> **Paper title revision and additional baseline comparison**
>
> First of all, we would like to thank you very much for your useful suggestions and very constructive reviews!
>
>
>
> **Regarding the comment:** "*Since the authors only tested the model on AMR graphs, they should mention it in the title. The approach is clearly general and it would be interesting see its application on other graph prediction tasks.*"
>
>
>
> **Answer:** even the implementation and experiments are mainly demonstrated for AMR graphs, the method that combines graph structure predictions is general, it enables further research on applying advanced ensembling methods like bagging, weighted voting etc. Moreover, the AMR graph is very general, it is a superset of dependency trees and other types of graphs. However, we are happy to revise the paper title to "Graph Ensemble Prediction for AMR Parsing".
>
>
>
> **Regarding the comment**: *"The main thing missing in this paper in my opinion is an ensemble baseline. For example, the transition base parser could be easily ensembled. Another type of ensembling could be stacking in sequence to sequence, where the output of a model is used as input for another model.  I understand that the way the authors proposed to merge graphs is novel, but I would also like to see a comparison with more "standard" ensembling approaches."*
>
>
>
> **Answer:** an important point to note is that this work is the first to study ensemble methods for combining AMR graphs rather than sequences. Before applying advanced ensembling algorithms like bagging (bootstrapping), weighted voting, etc. we need a technique that allows us to combine graph predictions. We focus on the development of this basic technique in this paper, as a starting point for the community to apply existing learning ensemble strategies. We leave this research direction opened as a future work as discussed in the conclusion section.
>
> In the paper, we have compared to a baseline denoted as Graphene 4S when 4 Spring models (at different checkpoints) are combined. Table 2 and 3 show that combining predictions from a diverse set of models yield better results than combining predictions from a single type of model with different checkpoints.
>
>
> Besides, we have implemented and compared the following baselines:
>
> -   **APT checkpoints ensemble**:  In Table 2 of the paper “AMR Parsing with Action-Pointer Transformer”, which we denote as APT, the authors report the scores from the ensemble that uses the average probability of 3 models from different seeds. The performance improvement ranges from 0.6 to 1.0 smatch point. When the baseline model performance is lower, i.e. 81.8, the improvement is 1.0. When the baseline model performance is higher, i.e. 82.8 (using 70k silver data), the improvement is 0.6 on AMR2.0. Our in-house experiments also indicate that as the baseline parser improves beyond 83.0, the impact of the probabilistic ensemble tends to decrease, consistent with the results reported in the OpenReview-anonymous paper “Structure-aware Fine-tuning of Sequence-to-sequence Transformers for Transition-based AMR Parsing”, where the authors report just 0.2 smatch improvement on AMR2.0 and 0.4 smatch improvement on AMR3.0 with the probabilistic ensemble technique in Table 1. On the other hand, as shown in Tables 2 and 3, our proposed technique achieves the performance gains of 1.3 -1.5 smatch points on the standard benchmark data sets and 0.9 - 1.8 on the out-of-distribution data sets, despite the fact that the highest performing system score is 84.1 on AMR2.0 and 83.1 on AMR3.0. Consistent high-performance gains across various datasets combined with the applicability to a diverse set of system outputs indicate the superiority of our proposed ensemble technique over other readily available model-specific ensemble methods.
> -   **Uniform sampling**:  for each set of predictions we sample the graph uniformly at random, this approach is equivalent to the “median” representative from a set.
> -   **Ideal median**: assumes that the gold AMRs are available for the test set (hence named as "ideal"). We computed the SMATCH of each prediction with the gold AMR and use the AMR with the median SMATCH score as the final prediction.
>
>   **Additional baselines comparison**
>
> |  | AMR 2.0 |AMR 3.0|BIO|New3|LP|
> |--|--|--|--|--|--|
> |  Uniform sampling| 0.824 | 0.828 | 0.560 | 0.711 | 0.779|
> |Ideal median | 0.835 | 0.836 | 0.577 | 0.731 | 0.788
>   Graphene | **0.858** | **0.844** | **0.622** | **0.757** | **0.804**

---

### Decision · Program_Chairs · 2021-09-27

**Decision:**

Accept (Poster)

**Comment:**

This paper details the problem of (labeled) graph ensembling, in particular for the task of semantic (AMR) parsing, and proposes a heuristic algorithm for solving it.

This is a challenging problem of interest. The reviewers find the work well-motivated and well explained, and raised some issues about comparisons, which the authors satisfactorily addressed with additional experiments in the discussion period. One outstanding point would be to clarify early on the scope of graphs for which the method works, more clearly. (However, the reviewers do not feel strongly that title should mention AMR, as the work is indeed more general.) Indeed, in some settings like link prediction, averaging of arc weights can be much simpler, and it might not be clear for all readers right away why and when the problem gets harder, based on the readers' background.

The reviewers made plenty of editing suggestions that I strongly encourage the authors to implement. In addition, I would like to add a few:

 - while I agree with the reviewer that it's better to rename the title of Section 3, the idiom provides helpful context so I would encourage you to keep it in the body (if not in the section title.)
 - line 83 and on: `$support$` should be styled as `$\operatorname{support}$` or defined as a macro with `$\DeclareMathOperator{\support}{support}$`.
 - line 146, the subscript `$g_{pivot}$` should be `$g_\text{pivot}$` -- same everywhere else.
 - all tables: try to align numbers on their decimal point (e.g. by right-alinging with a tabular font.)
 - Check the capitalization and formatting in your references. (e.g. "Amr" should be "AMR", "Machine learning" should be "Machine Learning" etc.)